# DreamBooth++: Boosting Subject-Driven Generation via Region-Level References Packing

### Zhongyi Fan[†]
Beijing Institute of Technology
Beijing, China
zyfanzy@foxmail.com

### Zixin Yin[†]
State Key Lab of Software
Development Environment,
Beihang University
Beijing, China
yzx835@buaa.edu.cn

### Gang Li
Institute of Software,
Chinese Academy of Sciences
University of Chinese Academy of
Sciences
Beijing, China
ucasligang@gmail.com

### Yibing Zhan
JD Explore Academy
Beijing, China
zhanyibing@jd.com

### Heliang Zheng[*]
University of Science and Technology
of China
Hefei, China
zhenghl@mail.ustc.edu.cn

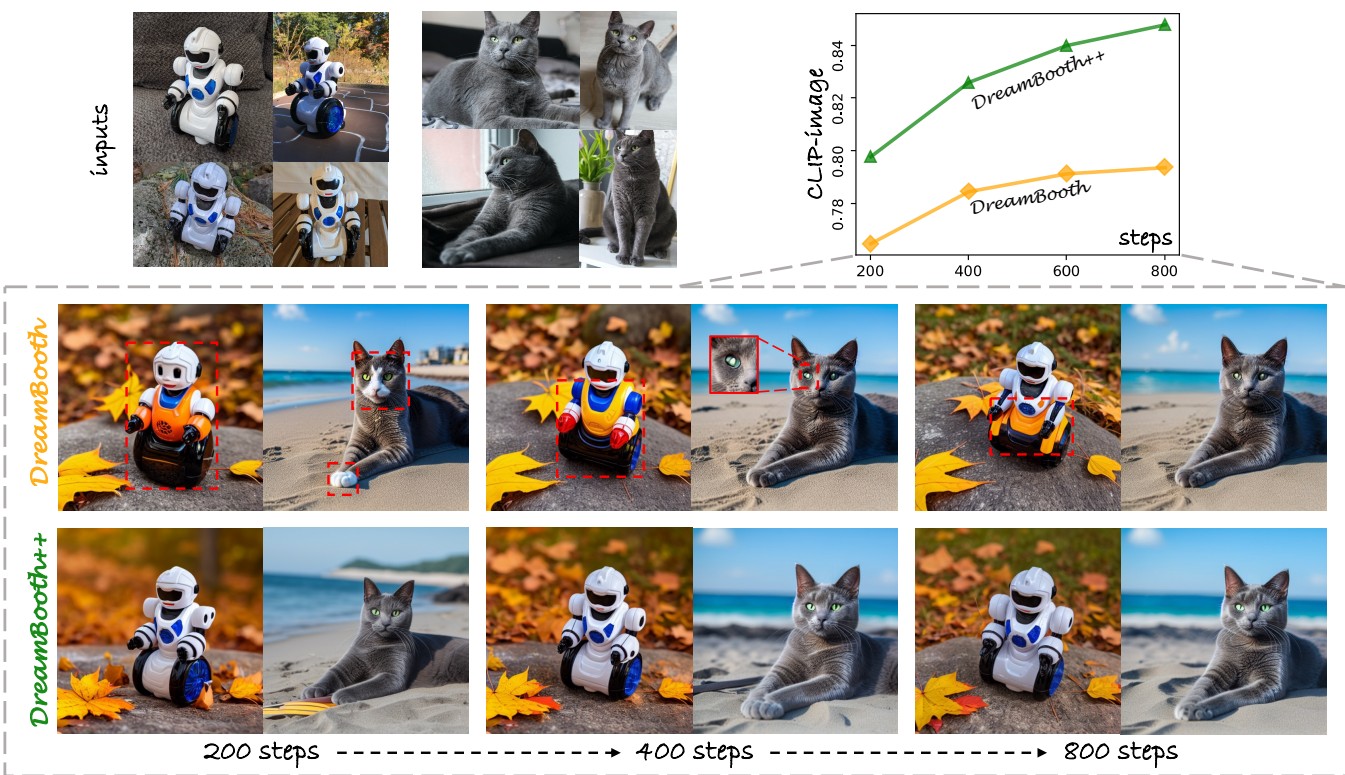

**Figure 1: Our proposed DreamBooth++ significantly improves the performance of DreamBooth with fewer training steps and higher subject fidelity. Prompts used in figures are "a [V] robot with fall leaves" and "a [V] cat on the beach".**

[†]Both authors contributed equally to this research.
[*]Corresponding author.

*MM '24, October 28-November 1, 2024, Melbourne, VIC, Australia*

## ABSTRACT

DreamBooth has demonstrated significant potential in subject-driven text-to-image generation, especially in scenarios requiring precise preservation of a subject's appearance. However, it still suffers from inefficiency and requires extensive iterative training to customize concepts using a small set of reference images. To address these issues, we introduce DreamBooth++, a region-level training strategy designed to significantly improve the efficiency and effectiveness of learning specific subjects. In particular, our approach employs a region-level data re-formulation technique that packs a set of reference images into a single sample, significantly reducing computational costs. Moreover, we adapt convolution and self-attention layers to ensure their processings are restricted within individual regions. Thus their operational scope (i.e., receptive field) can be preserved within a single subject, avoiding generating multiple sub-images within a single image. Last but not least, we design a text-guided prior regularization between our model and the pretrained one to preserve the original semantic generation ability. Comprehensive experiments demonstrate that our training strategy not only accelerates the subject-learning process but also significantly boosts fidelity to both subject and prompts in subject-driven generation.

## CCS CONCEPTS

• **Computing methodologies → Computer vision**.

## KEYWORDS

Subject-driven generation, Text-to-image synthesis, Diffusion model

**ACM Reference Format:**
Zhongyi Fan†, Zixin Yin†, Gang Li, Yibing Zhan, and Heliang Zheng*. 2024. DreamBooth++: Boosting Subject-Driven Generation via Region-Level References Packing. In *Proceedings of the 32nd ACM International Conference on Multimedia (MM '24), October 28-November 1, 2024, Melbourne, VIC, AustraliaProceedings of the 32nd ACM International Conference on Multimedia (MM'24), October 28-November 1, 2024, Melbourne, Australia.* ACM, New York, NY, USA, 10 pages. https://doi.org/10.1145/3664647.3680734

## 1 INTRODUCTION

Subject-driven text-to-image generation [9, 22, 27, 35] is an important application of diffusion-based generative models [18, 19, 25, 30, 31], which aims to accurately capture and reimagine the appearance of specific, previously unseen subjects from a small set of reference images. DreamBooth has emerged as a milestone in this domain, showing an impressive potential to maintain high fidelity in subject representation [27].

However, DreamBooth requires extensive iterative training and often struggles with slow convergence, imprecise detail preservation, and semantic mismatches [22, 40]. Some research [1, 5, 15, 20] focuses on getting rid of test-time tuning, a strategy that reduces computational costs and enhances flexibility. However, these approaches often fail to match the high fidelity of subject depiction provided by DreamBooth [36]. Thus designing an approach that can significantly reduce training time while precisely maintaining subject details turns out to be an urgent need.

Inspired by efficient example packing techniques such as patch 'n pack [6, 7], we present DreamBooth++, an adaptation of the DreamBooth methodology that introduces a region-level training strategy. This strategy shifts the focus to more granular, region-specific learning, enabling the model to learn the subject with impressive efficiency and accuracy. First, we present a region-level data re-formulation that redefines the input space of a diffusion model. Specifically, we partition the model inputs into multiple regions with different aspect ratios, each assigned an image of the subject. This approach allows our model to make the best use of computational resources by packing a set of reference images into a single sample. Further distinguishing DreamBooth++ are the adaptations we introduce to convolution and self-attention layers, alongside adjustments to the loss function. These adaptations are well designed to confine the operational scope of these layers within designated regions, thereby ensuring that the learning is region-specific. This strategic confinement is crucial for maintaining the original layout of images, effectively eliminating the issue of generating multiple sub-images within a single frame. Last but not least, we introduce a text-guided prior regularization to ensure that the cross-attention alignment between textual prompts and visual content in our fine-tuned model remains consistent with that of the pretrained model. This regularization is essential for maintaining semantic coherence between text and images, ensuring the generated images faithfully reflect the intended subjects and scenarios.

Note that our proposed region-level training strategy not only advances the efficient data packing re-formulation of the patch 'n pack [6] technique, adapting it for the 2D Unet [26] architecture, but also leverages the benefits of patch training [34]. Specifically, as shown in Figure 1, we experimentally find that our model (e.g., four-regions setting) surpasses the performance of DreamBooth even when the latter is optimized with four times as many iteration steps. Such an observation underscores the superiority of our method. Our contributions can be summarised as follows:

- We propose a novel data re-formulation and corresponding adaptations to model layers/loss for DreamBooth-based subject-driven generation. Such designs not only accelerate training convergence but also precisely capture subject details.
- We present a text-guided prior regularization that significantly enhances the semantic coherence between the text prompts and the visual outputs.
- We conduct extensive experiments to show that DreamBooth++ significantly speeds up the process of learning specific subjects and greatly enhances the details of the subject's appearance.

## 2 RELATED WORKS

### 2.1 Subject-Driven Generation

Diffusion-based text-to-image models [10, 18, 19, 21, 24, 25, 28, 30, 31] struggle with accurately rendering specific, unseen subjects. Subject-driven generation overcomes this by fine-tuning with novel subject images to synthesize ID-specific details. This method is divided into two categories based on whether fine-tuning is performed during the testing phase: fine-tuning based generation and fine-tuning free generation.

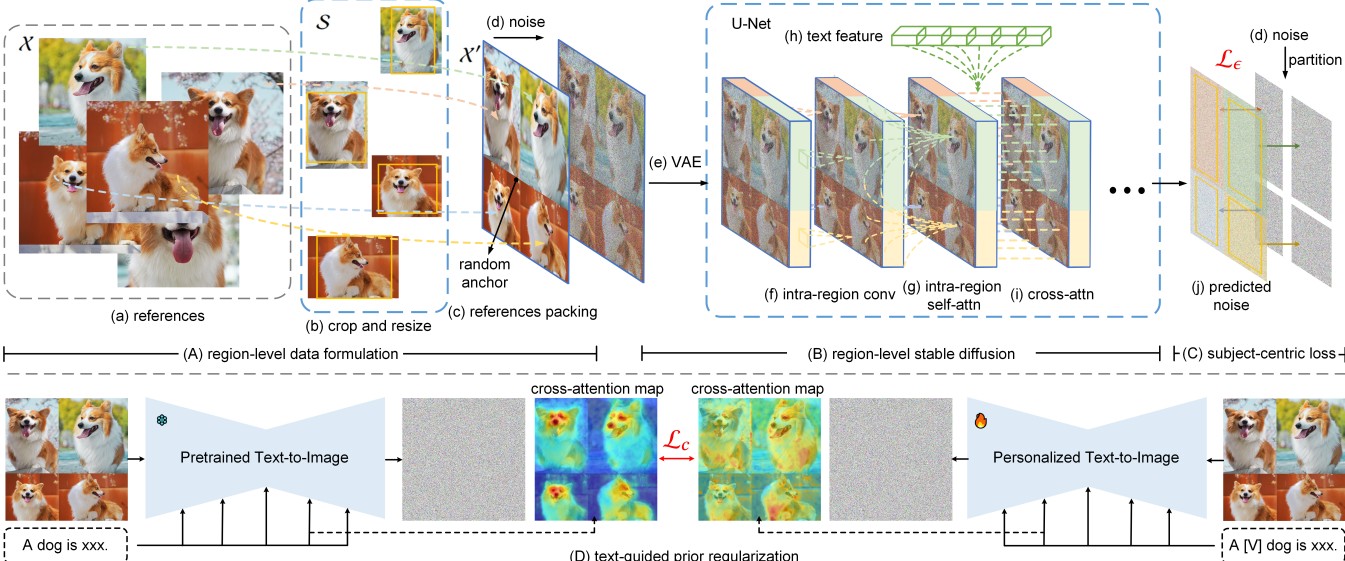

**Figure 2: An overview of our proposed DreamBooth++. (A) We reformulate the input space from the original training samples in (a) into packed samples in (c) by subject cropping and reference packing. (B) The packed samples are processed by an adapted U-Net architecture where the operational scope of convolution in (f) and self-attention operations in (g) are confined within each region, with parameters shared across different regions; (C) We only calculate loss on foreground areas to make it region-specific; (D) We optimize the cross-attention map of our model to align with that of the pretrained one, serving as a text-guided prior regularization.**

*Fine-Tuning Based Generation.* Fine-tuning based methods adapt pretrained text-to-image (T2I) models to new subjects by adjusting all or some parameters, such as those in the cross-attention layers. For instance, DreamBooth [27] employs fine-tuning using a customized token and a few subject images, applying reconstruction and class-specific prior preservation losses. Parameter-efficient fine-tuning approaches like LoRA [11] and OFT [22] aim to maintain appearance consistency with reduced computational demands, typically enhancing identity preservation but possibly restricting the model's generative capabilities. Inversion techniques such as Textural Inversion [9], ProSpect [39], and Celeb Basis [37] optimize text embeddings for special tokens to integrate specific IDs into the T2I model, preserving its overall generative capacity yet sometimes failing to capture unique subject characteristics accurately. Our DreamBooth++ framework extends the DreamBooth [27] methodology by incorporating novel data re-formulation techniques and carefully customized network structures with text-guided prior constraints. This not only improves the efficiency of learning with fewer steps but also improves the model's fidelity to text prompts and subject identities, facilitating dynamic and contextually appropriate image synthesis.

*Fine-Tuning Free Generation.* Alternatively, some approaches avoid fine-tuning during test time by employing image encoders or mapping networks to translate reference images into textual embeddings, guiding the generation process. Techniques like ELITE [35] and Instantbooth [29] utilize these networks to integrate image details into texts. Others, such as Taming Encoder [13] and SuTI [4], enhance semantic recognition and personalization, respectively.

Additionally, methods like IP-Adapter [36] focus on better feature separation, while MasaCtrl [2] and DreamTuner [12] add specialized attention modules for flexible synthesis without fine-tuning. These methods generally yield lower fidelity compared to fine-tuned models like DreamBooth [27] due to their reliance on weakly-supervised data and lack of adaptability during test time, which compromises quality and accuracy. In contrast, our DreamBooth++ leverages fine-tuning to achieve superior image quality and precise subject depiction essential for personalized image generation.

## 2.2 Example Packing

Example packing is a technique in Transformer-based models where multiple samples are concatenated into a single sequence, minimizing the need for padding and enhancing computational efficiency [7, 32]. This method accelerates language model training by efficiently handling variable-length inputs without cross-contamination [14]. In Vision Transformers (ViTs), a similar strategy called Patch n' Pack concatenates image patches to improve training efficiency [6, 8]. Patch Diffusion further optimizes this by segmenting image data to expedite training in diffusion models [34]. Our work introduces a "region-level" approach, which treats the VAE latent space as a canvas where different regions hold independent images, rather than representing a single, whole image as in traditional methods. Our adaptations further optimize example packing for 2D image space in subject-driven generation, aligning with mixed feature extraction methods of modern diffusion frameworks and enhancing overall efficiency.

# 3 METHOD

In this section, we introduce the framework of our proposed Dream-Booth++, an advanced training approach that refines the Dream-Booth [27] method by incorporating a region-level training strategy. Illustrated in Figure 2, DreamBooth++ addresses the challenges of slow convergence and semantic inconsistencies by focusing on three key aspects: 1) Region-level data re-formulation, which significantly improves the data efficiency by packing subject regions; 2) Region-level model adaptation, which applies intra-region convolution and self-attention operations, coupled with a subject-centric loss to ensure the learning process region-specific; and 3) text-guided prior regularization, which leverages cross-attention maps to preserve the pretrained model's rich semantic knowledge. We will introduce each component in detail.

## 3.1 Region-Level Data Re-formulation

Inspired by Patch 'n Pack [6] and Patch Diffusion [34], we propose a data re-formulation strategy that organizes multiple sub-images of a reference subject onto a single input canvas to enrich each training iteration with diverse representations of the subject. This approach not only enhances subject fidelity across all reference images but also accelerates the training process.

We initiate our region-level data re-formulation by defining the reference image set as: $\mathcal{X} = \{x^{(i)} \mid i = 1, 2, \ldots, n\}$, where each $x^{(i)}$ signifies a unique image within the set. Utilizing the state-of-the-art GroundingDINO [17] technique, we can accurately identify the subject's region of each image $x^{(i)}$, serving as the cornerstone for subsequent packing. This enables us to tailor cropping to any aspect ratio, ensuring each sub-image consistently preserves the subject's complete representation on the canvas.

*References Packing.* Given an empty layout grid that serves as the model's input canvas, we randomly create a grid of multiple regions using anchor points. This process is exemplified in Figure 2(c), where a single random anchor point divides the canvas into four distinct regions. We denote this set of regions as $\mathcal{R}$, with each region $r$ characterized by its aspect ratio and size, and the total number of regions is denoted by $k$. We then define the RegionAdaptiveResize function, which scales and crops the reference image $x$ to a specified region $r$, thereby maintaining the integrity of the subject throughout the process:

$$\mathcal{S} = \{s \mid s = \text{RegionAdaptiveResize}(x, r), x \in \mathcal{X}, r \in \mathcal{R}\}, \quad (1)$$

where $\mathcal{S}$ represents the collection of images that have been cropped and resized to fit designated regions on the canvas. This function is essential to adapting each image to fit its specific region on the canvas while maintaining the complete representation of the subject. Following the resizing, the Pack function then methodically arranges these cropped images to create the unified canvas input $x'$, which is designed to match the model's input dimensions. The Pack function is constructed according to the set of regions $\mathcal{R}$ on the canvas, aligning each cropped and scaled image to its assigned region $r$. Lastly, the set of inputs $\mathcal{X}'$, now richly diverse and precisely structured, emerges as:

$$\mathcal{X}' = \{x'^{(j)} \mid x'^{(j)} = \text{Pack}(\mathcal{R}_j, \mathcal{S}_j), j = 1, 2, \ldots, m\}, \quad (2)$$

where each $\mathcal{R}_j$ represents a unique canvas layout configuration determined by varying anchor placements, and $\mathcal{S}_j$ is a collection of images resized and adapted to fit the canvas layout. For an exemplar layout consisting of four sub-regions as shown in Figure 2(c), the spatial arrangement of the $\mathcal{S}$ is expressed as: $\begin{bmatrix} s_1 & s_2 \\ s_3 & s_4 \end{bmatrix}$.

This formulation optimizes the conventional data preparation in DreamBooth [27], yielding a significant improvement in both the effectiveness and efficiency of the model's training. This innovative data formulation ensures that each iteration during training exposes the model to a comprehensive view of the reference subject, fostering a deeper understanding and maintaining the fidelity of the subject's features.

## 3.2 Region-Level Model Adaptation

To address the challenges posed by our region-level data re-formulation, we modified the diffusion model's architecture to process each region independently. This change prevents the mixing of features between regions on the canvas, ensuring that each subject's identity is preserved distinctively. By applying a subject-centric loss to each isolated region, we enhance the model's focus and improve its ability to accurately capture and reproduce the unique characteristics of the subject.

*Intra-Region Convolution and Self-Attention Layers.* To maintain a distinct "single-subject receptive field" within the convolution and self-attention layers, we ensure that these layers process each subject region independently. This approach is crucial for mitigating the possibility of generating composite images, thus preserving the uniqueness of each subject region on the canvas.

The reformulated data sample $x'$, when propagated through the U-Net, reaches a specific layer where the input tensor is denoted as $\mathbf{z} \in \mathbb{R}^{h \times w \times d}$. Here, $h$, $w$, and $d$ represent the height, width, and feature dimension of the tensor, respectively. To respect the regional boundaries defined during the data re-formulation stage, $\mathbf{z}$ is partitioned into separate sections corresponding to the regions on the canvas:

$$\mathbf{z} = \bigoplus_{i=1}^{k} z_i, \quad (3)$$

where $\oplus$ signifies the concatenation of each tensor $z_i$, which corresponds to the regions defined by $\mathcal{R}$. Each convolution and self-attention operation is then confined to these individual regions to ensure localized processing:

$$\tilde{\mathbf{z}} = \bigoplus_{i=1}^{k} \mathrm{f}(z_i), \ \mathrm{f} \in \{\text{Conv}, \text{Self-Attn}\}. \quad (4)$$

Here, $\mathrm{f}(\cdot)$ represents either the standard convolution or standard self-attention operation. This strategy guarantees that each region is processed separately, preventing the mixing of features across regions in the output tensor $\tilde{\mathbf{z}}$. This approach mitigates potential issues like multiple subjects from our reference packing strategy by isolating interactions between regions. For cross-attention layers that focus on text-image correspondence and share the same prompt across all regions, no alterations are needed as they naturally do not interfere with regional boundaries.

*Subject-Centric Loss.* We utilize GroundingDINO[17] to identify the subject areas on $x'$ and create a corresponding mask $M \in \mathbb{R}^{h \times w}$. The modified diffusion loss is thus defined:

$$\mathcal{L}_\epsilon = \left\| \epsilon_\theta(\alpha_t x' + \beta_t \epsilon, \mathbf{c}, t) - \epsilon \right\|_2^2 \odot M, \tag{5}$$

where $t \in [0, 1]$ indicates a time step randomly selected from the diffusion process, $\theta$ denotes the parameters of the network, $\epsilon \in \mathbb{R}^{h \times w}$ represents the Gaussian noise, and $\alpha_t$ and $\beta_t$ are coefficients derived from the noise schedule. This element-wise multiplication with the mask $M$ not only enhances precision but also streamlines the learning process, ensuring the model efficiently captures subject characteristics with high fidelity.

## 3.3 Text-Guided Prior Regularization

In subject-driven generation, it's crucial for the model to link a subject with a specific identifier token, enabling the manipulation of subject attributes through textual prompts. However, the fine-tuning process of DreamBooth [27] often results in reduced prompt fidelity due to the model's overfitting to the limited context of reference images. This can cause semantic drift [27], where the subject-specific token incorrectly aligns with unrelated contexts.

To mitigate semantic drift, DreamBooth++ employs a regularization strategy on the cross-attention layers, essential for aligning textual prompts with visual content. This approach, inspired by [16], maintains the rich semantic context of the pretrained model. Specifically, we define $\mathbf{c}$ as the conditioning vector from the text prompt "an image of [V]". We replace the placeholder"[V]" with the specific class name, resulting in $\mathbf{c}_r$(e.g., "an image of [class name]"). This adjustment facilitates the application of a text-guided prior regularization loss, which is represented by the equation:

$$\mathcal{L}_c = \left\| CA_\theta(x', \mathbf{c}) - CA_r(x', \mathbf{c}_r) \right\|_2^2, \tag{6}$$

where $CA_\theta(x', \mathbf{c})$ and $CA_r(x', \mathbf{c}_r)$ represent the cross-attention maps of the fine-tuned and pretrained models, respectively. This regularization not only confines the influence of the specialized identifier to the subject but also ensures that the attention patterns for other tokens align with those of the pretrained model, facilitating comprehensive semantic consistency essential for accurate subject-driven generation.

## 3.4 Comprehensive Optimization Objective

Following DreamBooth [27], we also introduce the prior-preservation loss as

$$\mathcal{L}_{pr} = \left\| \epsilon_\theta(\alpha_t x_{pr} + \beta_t \epsilon, \mathbf{c}_{pr}, t) - \epsilon \right\|_2^2, \tag{7}$$

where the class-specific prior samples $\{x_{\mathrm{pr}}^{(1)}, x_{\mathrm{pr}}^{(2)}, ...\}$ is sampled from the frozen pretrained text-to-image diffusion model with conditioning prompt $c_{\mathrm{pr}} \coloneqq$ "a photo of [class name]".

The overall optimization objective of DreamBooth++ combines the subject-centric loss $\mathcal{L}_\epsilon$, the prior-preservation loss $\mathcal{L}_{pr}$, and the text-guided prior regularization loss $\mathcal{L}_c$, formulated as:

$$\mathcal{L} = \mathcal{L}_\epsilon + \mathcal{L}_{pr} + \lambda \mathcal{L}_c. \tag{8}$$

where $\lambda$ is a weighting factor adjusting the influence of the text-guided prior regularization. Through this comprehensive optimization framework, DreamBooth++ not only captures subject details

**Table 1: Quantitative comparison of subject fidelity (DINO, CLIP-I), prompt fidelity (CLIP-T), and diversity (LPIPS) across methods, alongside convergence steps (C-steps). *Pretrained* shows the model without fine-tuning, setting a theoretical upper limit for CLIP-T, while *Real Images* provides reference metrics for DINO and CLIP-I by measuring similarity among reference images without self-correlation.**

| Methods | DINO↑ | CLIP-I↑ | CLIP-T↑ | LPIPS↑ | C-steps↓ |
|---|---|---|---|---|---|
| *Pretrained* | 0.320 | 0.643 | 0.267 | 0.846 | - |
| *Real Images* | 0.711 | 0.857 | - | 0.688 | - |
| TI [9] | 0.564 | 0.739 | 0.213 | **0.814** | 4000 |
| DB [27] | 0.642 | 0.794 | 0.236 | 0.736 | 1000 |
| LoRA [11] | 0.637 | 0.792 | 0.239 | 0.751 | 1000 |
| OFT [22] | 0.670 | 0.802 | 0.234 | 0.736 | 1400 |
| DB++ (Ours) | **0.673** | **0.826** | **0.242** | 0.750 | **400** |

precisely but also responds flexibly to various text prompts, generating images that maintain semantic consistency and accurately represent the subject's appearance.

## 4 EXPERIMENTS

### 4.1 Experiment Setup

*4.1.1 Compared Methods.* We conduct comparative analyses of DreamBooth++ (DB++) against several fine-tuning-based methods such as DreamBooth (DB) [27], Dreambooth-LoRA (LoRA) [11], OFT [22], and Textural Inversion (TI) [9]. For DreamBooth, LoRA, and Textural Inversion, we utilize the open-source code from the diffusers library [33]. For OFT, we employ the official implementation [1] for evaluations. In all experiments, Stable Diffusion v1.5 [25] serve as our pretrained text-to-image diffusion model, tested on the Dreambooth dataset [27]. We adhered to the recommended hyperparameters specified in the respective publications. Each of the DreamBooth-based methods was trained using generated images as regularization. Regarding model checkpoint selection, we fine-tune each method for an adequate period and report the results from the first checkpoint that demonstrated optimal performance.

*4.1.2 Implementation Details.* Our experiments utilize the Dreambooth implementation from the diffusers library [33]. The hyperparameters remain largely consistent with those specified for Dream-Booth, except where explicitly noted. For the Dreambooth dataset, each subject is represented in 25 to 35 unique layouts, with original images evenly distributed across canvases configured in either $k = 4$ (2x2) or $k = 9$ (3x3) layouts. The learning rate during the training phase is consistently set at $2e - 5$. Additionally, the weight for text-guided prior regularization, $\lambda$, is fixed at 10 to balance textual fidelity and visual accuracy in the generated images.

*4.1.3 Evaluation Metrics.* Adopting the evaluation framework from DreamBooth [27], we utilize a comprehensive set of metrics to assess each method's performance. These metrics include the average

---
[1]https://github.com/Zeju1997/oft

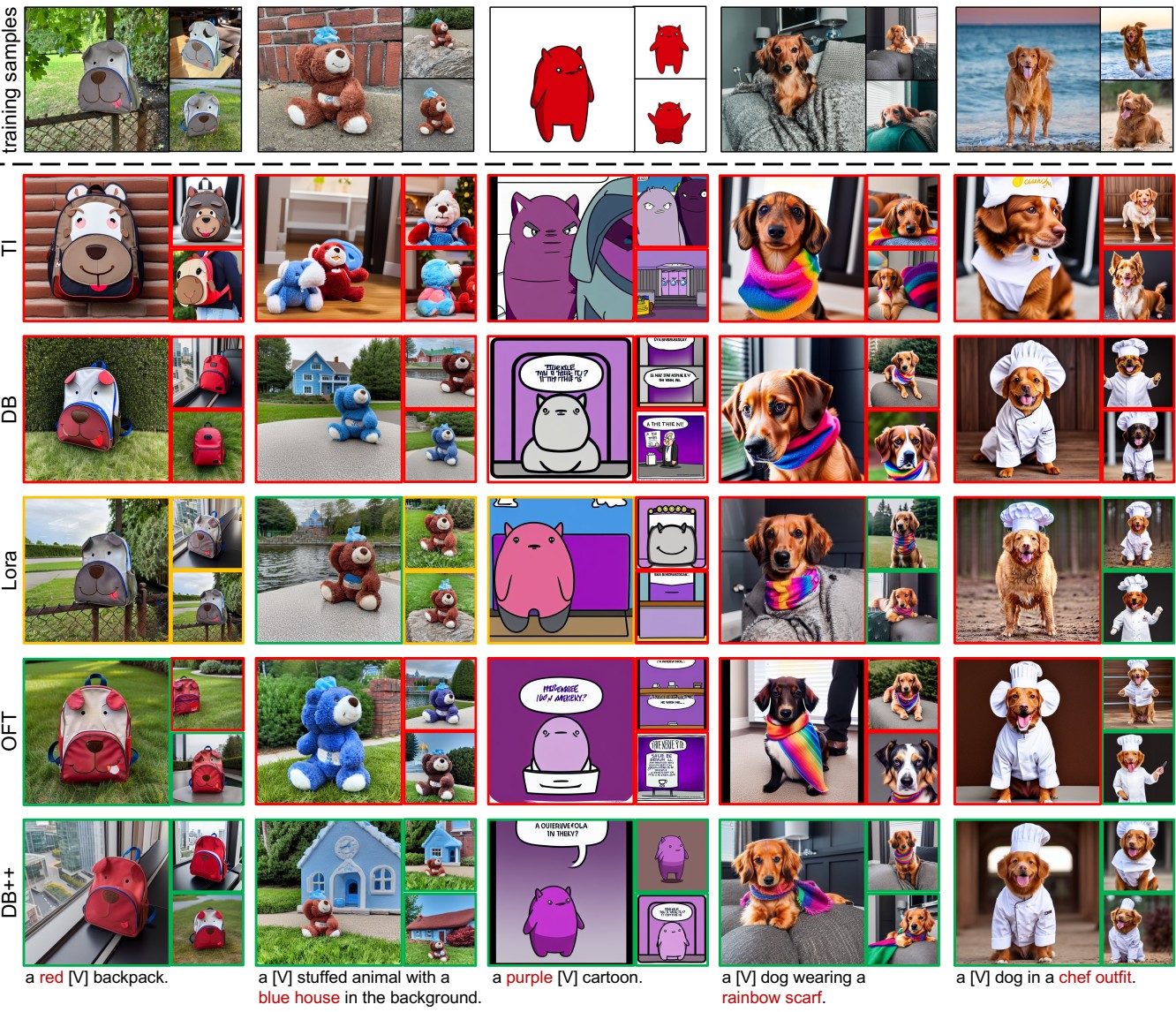

**Figure 3: Qualitative comparison of subject-driven generation among different methods, including Dreambooth (DB) [27], LoRA) [11], Text Inversion (TI) [9], OFT [22], and Dreambooth++ (DB++). In the results, green borders signify high fidelity in both semantics and appearance; red borders indicate issues with appearance; and yellow borders emphasize semantic inconsistencies.**

cosine similarity between CLIP image embeddings and text embeddings (CLIP-T [23]) to evaluate the alignment between generated images and their corresponding text prompts, and the average cosine similarity between CLIP image embeddings and reference images (CLIP-I [23]) to measure subject fidelity. Additionally, we use DINO embeddings [3] to precisely evaluate unique subject features, ensuring detailed fidelity between generated and reference images. The Local Perceptual Image Patch Similarity (LPIPS)[38] metric is used to assess the diversity of images by calculating the average cosine similarity between multiple images generated from

the same subject and prompt. We also include the convergence steps (C-steps) for each method, providing a quantitative measure of the efficiency of our approach.

## 4.2 Quality Comparison with Other Methods

We conduct a comprehensive comparison of DreamBooth++ with other leading subject-driven image generation methods, including DreamBooth, Textual Inversion, LoRA, and OFT. The results are depicted in Figure 3 and detailed quantitative metrics are presented in Table 1. This table includes benchmarks such as *Pretrained*, which

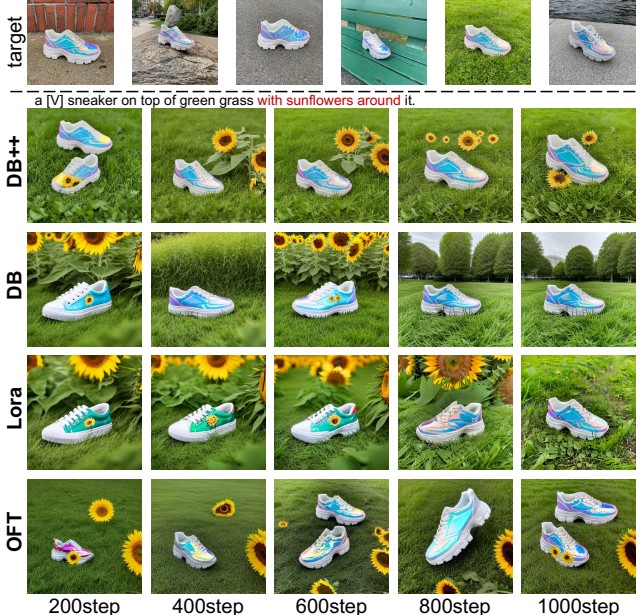

**Figure 4: Convergence analysis comparing DreamBooth++ with other methods including (DB) [27], LoRA) [11], and OFT [22] over different iteration counts.**

represents the model's performance without fine-tuning, providing a theoretical upper limit for CLIP-T fidelity, and *Real Images*, which measures the similarity among reference images without self-correlation to establish reference values for DINO and CLIP-I metrics. If metric values exceed these reference levels, it can be considered a sign of severe overfitting. The experimental findings emphatically demonstrate that DreamBooth++ not only surpasses all other fine-tuning-based methods but achieves so with fewer iterations.

DreamBooth++ excels in generating images with richer detail, higher fidelity to the subjects, and closer adherence to the semantics of the prompts. This efficiency in convergence, coupled with its ability to produce high-quality, subject-specific images that are visually accurate and textually coherent, highlights DreamBooth++'s effectiveness in subject-driven image generation tasks. It showcases superior performance in generating images that not only faithfully reflect the visual and contextual subtleties of their subjects but also do so more swiftly and efficiently than existing methods.

### 4.3 Fine-tuning Convergence

We assess the performance of DreamBooth++, DreamBooth, LoRA, and OFT by comparing their effectiveness at different numbers of iterations. Given that Text Inversion focuses on optimizing textual embeddings and has a notably slow convergence rate, we excluded it from this comparison. As depicted in Figure 4, DreamBooth++ achieves high consistency in identity appearance after approximately 400 iterations, significantly reducing the resource costs associated with fine-tuning diffusion models. In contrast, methods

**Table 2: Ablation study on each component of our proposed Dreambooth++ at 400 training steps.**

| Methods | DINO↑ | CLIP-I↑ | CLIP-T↑ | LPIPS↑ |
|---|---|---|---|---|
| Baseline (400 steps) | 0.621 | 0.784 | 0.242 | 0.749 |
| + References Packing | 0.624 | 0.783 | 0.240 | **0.782** |
| + Model Adaptation | **0.686** | **0.833** | 0.229 | 0.735 |
| + Text-guided Prior (Full) | 0.673 | 0.826 | **0.242** | 0.750 |

such as DreamBooth and LoRA requires more than twice the number of iterations to reach comparable outcomes.

From our analysis in Figure 4, we observe several key trends in the behavior of different models over various iterations: (1) *Initial Confusion*: During the initial phases, models often mishandle the semantic features between disparate subjects such as "sunflowers" and "sneakers," leading to incorrect semantic blending. DreamBooth++ quickly corrects these incorrect attention patterns through its text-guided prior regularization, ensuring both textual consistency and high fidelity in subject identity. (2) *Tendency to Overfit*: As Dream-Booth and LoRA further refine their details of subject, they typically overfit to specific subjects, which leads to a loss in the ability to preserve prompt semantics, resulting in a noticeable decline in generated quality. (3) *Consistent Semantic Preservation*: OFT stands out due to its robust semantic preservation capabilities, maintaining accurate prompt semantics consistently across iterations. However, its slower convergence rate and the strict requirements for semantic preservation somewhat limit its ability to precisely describe to the given subject. These findings underscore the effectiveness and stable convergence of our approach in subject-driven generation, demonstrating DreamBooth++ as a superior method in terms of both efficiency and fidelity.

### 4.4 Ablation Study

We conduct an ablation study to evaluate the impact of various enhancements in DreamBooth++. This analysis is aimed at understanding the isolated and combined effects of each modification on the model's performance. The results, illustrated in Figure 6 and quantified in Table 2, not only highlight the value of each component but also demonstrate their collective impact in improving fidelity and efficiency.

*Effect of Region-Level Data Re-formulation.* Our integration of region-level data re-formulation shows mixed effects on performance metrics, as observed in the outcomes detailed in Table 2 and illustrated in Figure 6. The DINO metric slightly improved, suggesting enhanced subject fidelity, while the CLIP-T score decreased, indicating reduced alignment with textual prompts. The visual disturbances such as the appearance of multiple subjects or visible seams in generated images, highlighted in the second column of Figure 6, exemplify the artifacts introduced by this component. These artifacts not only affect the aesthetic quality of the images but also contribute to the inconsistency in output. This suggests that while the model's efficiency in capturing the appearance of subjects improved, the introduction of artifacts undermined the overall quality and coherence of the generated images.

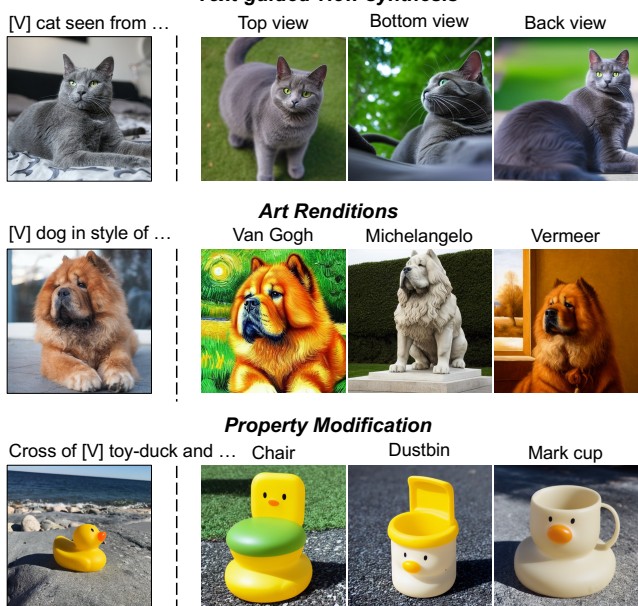

**Figure 5: DreamBooth++ versatility through text-guided view synthesis, art renditions, and property modifications. For each subject, different contexts and artistic styles are synthesized, and attributes such as apparel are altered while maintaining consistency in image and text fidelity.**

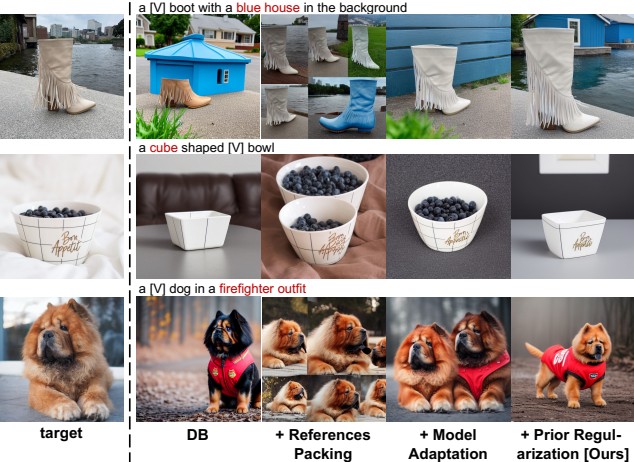

**Figure 6: Ablation study visualizing the impact of different components: (a) baseline (DreamBooth); (b) DreamBooth with data re-formulation; (c) DreamBooth++ without text-guided prior regularization; (d) full DreamBooth++ implementation.**

*Effect of Region-Level Model Adaptation.* The third column of Figure 6 illustrates how the regional isolation impacts the model's performance. While this adaptation significantly accelerates the model's ability to capture the specified subjects' appearance, it also

introduces challenges in preserving prompt semantics. Specifically, the model tends to overlook elements in the prompt that are unrelated to the main subject (e.g., rugs, houses) and alterations to subject attributes (e.g., changing colors to blue, shapes to cubes). These examples underscore the modification's dual impact: enhancing subject detail capture at the expense of reducing semantic fidelity to the text prompts.

*Effect of Text-guided Prior Regularization.* As evidenced in the final column of Figure 6, the application of text-guided prior regularization notably corrects semantic inconsistencies, a common issue in models suffering from overfitting. The introduction of this regularization is reflected quantitatively in our metrics, where there is a noticeable improvement in CLIP-T scores by approximately 0.013, as detailed in Table 2. However, this semantic enhancement compromises the model's ability to maintain subject appearance, leading to a minor decline in both DINO and CLIP-I metrics. Despite this, the overall visual quality of the images is elevated, ensuring a consistent depiction that spans both image details and textual semantics. This balancing act between visual and semantic fidelity highlights the effectiveness of our regularization approach.

## 4.5 Subject-driven Applications

Figure 5 demonstrates DreamBooth++'s versatility across subject-driven applications like text-guided view synthesis, art renditions, and property modification. The examples highlight DreamBooth++'s proficiency in preserving subject identity across diverse semantic contexts. Notably, the model adeptly incorporates subjects into various artistic styles without a sense of incongruity, as seen in its successful emulation of Van Gogh's distinctive brushstroke textures. Furthermore, in scenarios involving property modifications, DreamBooth++ creatively blends the appearance of a toy duck with different object properties, demonstrating its capability to generate inventive images. This adaptability confirms DreamBooth++'s effectiveness in generating high-fidelity images that accurately reflect the specified prompts while adapting seamlessly to various environments and artistic influences.

## 5 CONCLUSION

This paper introduces DreamBooth++, an enhancement of the DreamBooth methodology that significantly improves subject-driven image generation. By integrating a region-level data re-formulation, model adaptation, and text-guided prior regularization, DreamBooth++ boosts computational efficiency and ensures superior fidelity in subject representation while enhancing semantic consistency. Our comprehensive experiments demonstrate that DreamBooth++ surpasses existing methods, offering a robust solution for producing high-quality, subject-specific images efficiently. In the future, we will combine this framework with other efficient fine-tuning approaches to further enhance the quality and efficiency of subject-driven generation. Additionally, our method's flexibility in handling images of arbitrary aspect ratios could significantly improve data utilization during the pre-training phase, potentially offering substantial advancements in model training methodologies.

## ACKNOWLEDGMENTS

This work was partially supported by the Major Science and Technology Innovation 2030 "New Generation Artificial Intelligence" key project (No. 2021ZD0111700).

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
