# OpenReview forum: "DreamBooth++: Boosting Subject-Driven Generation via Region-Level References Packing"
_acmmm.org/ACMMM/2024/Conference — MM2024 Poster_

### Official Review · Reviewer_Chhc · 2024-05-22

**Rating:** 3
**Confidence:** 3

**Summary:**

This paper introduces a new technique for DreamBooth, grounded in subject-driven generation. The proposed training strategy is both intriguing and straightforward. Based on the current version of the manuscript, I offer the following comments:

1. To further evaluate the proposed method, additional metrics for assessing visual authenticity should be included, such as the Fréchet Inception Distance (FID) and the Inception Score.

2. The author(s) should explicitly detail the training environments utilized for the proposed method, with specific emphasis on the GPU specifications and batch size employed.

3. For a generation task of this nature, conducting a user study would be advantageous to assess practical effectiveness and user satisfaction.

4. In Table 2, the full model does not achieve the best performance compared to other variants. The author(s) should highlight the highest scores in bold rather than those of the full model. Additionally, more explanations should be provided regarding this observation. It is noted that CLIP-I and CLIP-T exhibit opposite behaviors. This raises questions about the effectiveness of the proposed method and the metrics adopted. Further experiments should be conducted to validate these findings.

**Strengths:**

This paper introduces a new technique for DreamBooth, grounded in subject-driven generation. The proposed training strategy is both intriguing and straightforward.

**Limitations:**

1. To further evaluate the proposed method, additional metrics for assessing visual authenticity should be included, such as the Fréchet Inception Distance (FID) and the Inception Score.

2. The author(s) should explicitly detail the training environments utilized for the proposed method, with specific emphasis on the GPU specifications and batch size employed.

3. For a generation task of this nature, conducting a user study would be advantageous to assess practical effectiveness and user satisfaction.

4. In Table 2, the full model does not achieve the best performance compared to other variants. The author(s) should highlight the highest scores in bold rather than those of the full model. Additionally, more explanations should be provided regarding this observation. It is noted that CLIP-I and CLIP-T exhibit opposite behaviors. This raises questions about the effectiveness of the proposed method and the metrics adopted. Further experiments should be conducted to validate these findings.

**Suitability:**

3

---

### Official Review · Reviewer_sRH8 · 2024-05-24

**Rating:** 5
**Confidence:** 2

**Summary:**

This paper focuses on improvements of the DreamBooth in terms of efficiency and precise preservation of a subject’s appearance, which introduces DreamBooth++, a region-level training strategy designed to significantly improve the efficiency and effectiveness of learning specific subjects. Comprehensive experiments demonstrate that their training strategy not only accelerates the subject-learning process but also significantly boosts fidelity to both subject and prompts in subject-driven generation.

**Strengths:**

[+] The quality of the paper is good in terms of novelties and analysis.

[+] The writing is generally clear with concise implementation details and a detailed description of the training pipeline.

**Limitations:**

[-] Please further optimize the layout of the paper, e.g. Table 2.
[-] My main concern lies in the experiments: (1) The existing experiments, including comparison and ablation experiments, do not allow me to discover intuitively the impact that the proposed method brings to the efficiency. It would be nice to have comparison curves with other methods for different rounds of training. (2) The increase in efficiency comes from the use of which technology, which was not demonstrated in the ablation experiments. References Packing or Text-guided Prior?
[-] I'm not sure if region-level data reformulation will bring huge costs in the training phase, e.g. GPU memory usage. It would be nice to discuss this trade-off.

**Suitability:**

2

---

### Official Review · Reviewer_Wcho · 2024-05-26

**Rating:** 3
**Confidence:** 4

**Summary:**

The paper introduces DreamBooth++, an innovative approach to subject-driven image generation that enhances the original DreamBooth methodology. The authors focus on a region-level training strategy that efficiently encapsulates the distinctive features of specific subjects using a novel data re-formulation technique. This technique involves packing multiple reference images into a single sample, which is particularly beneficial for computational efficiency and the acceleration of the learning process. The paper claims that DreamBooth++ not only quickens the subject-learning process but also enhances the fidelity of the generated images to both the subject and the textual prompts provided.

**Strengths:**

1、The paper proposes a pioneering method for subject-specific image generation that could potentially set a new standard in the field.

2、The authors have presented their work in a manner that is easy to follow, with clear explanations and well-structured content.

3、The paper presents a balanced view with both qualitative examples and quantitative metrics, offering a comprehensive understanding of the method's performance.

**Limitations:**

1、To the best of the reviewers' knowledge, many current personalized generation methods focus on a single reference image. However, the region-level data re-formulation proposed in this paper appears to face challenges when applied to a single reference image.

2、There is an absence of comparison with existing data augmentation techniques, which is crucial for establishing the superiority of the proposed method.

3、Despite mentioning improved training efficiency, the paper does not provide concrete data on training time, which is a critical metric for evaluating the practicality of the method.

4、Presenting results for only a single subject limits the generalizability of the findings. Including a variety of subjects would strengthen the paper's conclusions.

5、Some expressions used in the paper could be more precise. For instance, the exact nature of the 'region-level' enhancements and how they differ from existing techniques should be clarified.

6、Since the Code is not given, the paper would benefit from more detailed experimental setup and hyperparameter settings to allow for reproducibility of the results by other researchers.

**Suitability:**

3

---

### Official Review · Reviewer_Qayq · 2024-05-31

**Rating:** 4
**Confidence:** 2

**Summary:**

This paper proposed DreamBooth++ to deal with subject-driven text to image generation. A region-level subject adaptation mechanism was proposed, based on an idea similar to Patch n’Pack, for efficient adaptation and faster convergence. An additional text-guided prior regularization term was proposed to minimize the difference between cross-attention maps of the adaptation model and a frozen pre-trained model. Experimental results on the DreamBooth datasets demonstrated improved metrics compared to existing adaptation approaches.

**Strengths:**

-	This paper is well-motivated, clearly written, and easy to follow.
-	I like the conceptual simplicity of the proposed approach: subject-centric loss to encourage fidelity, cross-attention map regularization to encourage semantic consistency with the prompt. Each new component seems to be necessary and complementary to each other.
-	Experimental results showed that this approach has achieved improved results both quantitatively and qualitatively.

**Limitations:**

-	More proof is needed to support the ‘speed up’ claims: Although multiple images are packed together into a single training instance, computation cost might have increased due to the additional regularization term. It would be good if the authors could provide more information on convergence speed in terms of time consumption in addition to convergence iterations.
-	Although the region-level adaptation is conceptually simple, I imagine the implementation of the localized processing operations (e.g., Eq(4)) could possibly involve quite a bit effort, which might make it challenging for other researchers to reproduce.
-	Lacking hyperparameter sensitivity analysis: $\lambda$ controls the weight of regularization term, and it’s not clear how sensitive this approach is. It is fixed to $10$ in all the experiments. Since there is only one dataset in this study, it might not be the best value to generalize to other datasets.
-	Grounding-DINO was used to detect subject and provide masks in the adaptation loss (Eq(5)). Would it make the comparisons less fair? Could it be used in other approaches to produce stronger baselines?

Minor:

-	It would be good if the authors could check some failure cases.
-	Table 2 has overflow issues and the last column was partially occluded by Figure 5.

Overall, I think this is a good paper. I’d like to hear the authors’ feedback and other reviewers’ comments.

**Suitability:**

3

---

### Meta-Review · Area_Chair_8qrs · 2024-06-30

**Recommendation:** Accept (Poster)
**Confidence:** 5

**Metareview:**

Three reviewers rate borderline accept, and one reviewer rates borderline reject, with the main concern on small-scale evaluation. Despite some concerns on the evaluation, reviewers all acknowledge the contribution of the work. Area chair would recommend accept as poster and encourage the authors to take the reviewers' comments into considerations when preparing for the camera-ready version.